# FILLING THE G_AP_S: MULTIVARIATE TIME SERIES IMPUTATION BY GRAPH NEURAL NETWORKS

**Andrea Cini**[*1]**, Ivan Marisca**[*1]**, Cesare Alippi**[12]
[1]The Swiss AI Lab IDSIA, Università della Svizzera italiana [2]Politecnico di Milano

## ABSTRACT

Dealing with missing values and incomplete time series is a labor-intensive, tedious, inevitable task when handling data coming from real-world applications. Effective spatio-temporal representations would allow imputation methods to reconstruct missing temporal data by exploiting information coming from sensors at different locations. However, standard methods fall short in capturing the nonlinear time and space dependencies existing within networks of interconnected sensors and do not take full advantage of the available – and often strong – relational information. Notably, most state-of-the-art imputation methods based on deep learning do not explicitly model relational aspects and, in any case, do not exploit processing frameworks able to adequately represent structured spatio-temporal data. Conversely, graph neural networks have recently surged in popularity as both expressive and scalable tools for processing sequential data with relational inductive biases. In this work, we present the first assessment of graph neural networks in the context of multivariate time series imputation. In particular, we introduce a novel graph neural network architecture, named `GRIN`, which aims at reconstructing missing data in the different channels of a multivariate time series by learning spatio-temporal representations through message passing. Empirical results show that our model outperforms state-of-the-art methods in the imputation task on relevant real-world benchmarks with mean absolute error improvements often higher than $20\%$.

## 1 INTRODUCTION

Imputation of missing values is a prominent problem in multivariate time-series analysis (TSA) from both theoretical and practical perspectives (Little & Rubin, 2019). In fact, in a world of complex interconnected systems such as those characterizing sensor networks or the Internet of Things, faulty sensors and network failures are widespread phenomena that cause disruptions in the data acquisition process. Luckily, failures of these types are often sparse and localized at the single sensor level, i.e., they do not compromise the entire sensor network at once. In other terms, it is often the case that, at a certain time step, missing data appear only at some of the channels of the resulting multivariate time series. In this context, spatio-temporal imputation methods (Yi et al., 2016; Yoon et al., 2018b) aim at reconstructing the missing parts of the signals by possibly exploiting both temporal and spatial dependencies. In particular, effective spatio-temporal approaches would reconstruct missing values by taking into account past and future values, and the concurrent measurements of spatially close neighboring sensors too. Here, *spatial similarity* does not necessarily mean physical (e.g., geographic) proximity, but rather indicates that considered sensors are related w.r.t. a generic (quantifiable) functional dependency (e.g., Pearson correlation or Granger causality – Granger, 1969) and/or that are close in a certain latent space. Relational information, then, can be interpreted as a set of constraints – linking the different time series – that allows replacing the malfunctioning sensors with virtual ones.

Among different imputation methods, approaches based on deep learning (LeCun et al., 2015; Schmidhuber, 2015; Goodfellow et al., 2016) have become increasingly popular (Yoon et al., 2018a; Cao et al., 2018; Liu et al., 2019). However, these methods often completely disregard available relational information or rely on rather simplistic modifications of standard neural architectures tailored

---

[*]Equal contribution. Correspondence to `andrea.cini@usi.ch`.

for sequence processing (Hochreiter & Schmidhuber, 1997; Chung et al., 2014; Bai et al., 2018; Vaswani et al., 2017). We argue that stronger, structural, inductive biases are needed to advance the state of the art in time series imputation and allow to build effective inference engines in the context of large and complex sensor networks as those found in real-world applications.

In this work, we model input multivariate time series as sequences of graphs where edges represent relationships among different channels. We propose graph neural networks (GNNs) (Scarselli et al., 2008; Bronstein et al., 2017; Battaglia et al., 2018) as the building block of a novel, bidirectional, recurrent neural network for multivariate time series imputation (MTSI). Our method, named *Graph Recurrent Imputation Network* (`GRIN`), has at its core a recurrent neural network cell where gates are implemented by message-passing neural networks (MPNNs; Gilmer et al., 2017). Two of these networks process the input multivariate time series in both forward and backward time directions at each node, while hidden states are processed by a message-passing imputation layer which is constrained to learn how to perform imputation by looking at neighboring nodes. In fact, by considering each edge as a soft functional dependency that constraints the value observed at corresponding nodes, we argue that operating in the context of graphs introduces a positive inductive bias for MTSI. Our contributions are manifold: 1) we introduce a methodological framework to exploit graph neural networks in the context of MTSI, 2) we propose a novel, practical and effective implementation of a GNN-based architecture for MTSI, and 3) we achieve state-of-the-art results on several and varied MTSI benchmarks. Our method does not rely on any assumption on the distribution of the missing values (e.g., presence and duration of transient dynamics and/or length of missing sequences) other than stationarity of the underlying process. The rest of the paper is organized as follows. In Section 2 we discuss the related works. Then, in Section 3, we formally introduce the problem settings and the task of MTSI. We present our approach to MTSI in Section 4, by describing the novel framework to implement imputation architectures based on GNNs. We proceed with an empirical evaluation of the presented method against state-of-the-art baselines in Section 5 and, finally, we draw our conclusions in Section 6.

## 2 RELATED WORKS

**Time series imputation**   There exists a large literature addressing missing value imputation in time series. Besides the simple and standard interpolation methods based on polynomial curve fitting, popular approaches aim at filling up missing values by taking advantage of standard forecasting methods and similarities among time series. For example, several approaches rely on k-nearest neighbors (Troyanskaya et al., 2001; Beretta & Santaniello, 2016), the *expectation-maximization* algorithm (Ghahramani & Jordan, 1994; Nelwamondo et al., 2007) or linear predictors and state-space models (Durbin & Koopman, 2012; Kihoro et al., 2013). Low-rank approximation methods, such as matrix factorization (Cichocki & Phan, 2009), are also popular alternatives which can also account for spatial (Cai et al., 2010; Rao et al., 2015) and temporal (Yu et al., 2016; Mei et al., 2017) information. Among linear methods, STMVL (Yi et al., 2016) combines temporal and spatial interpolation to fill missing values in geographically tagged time series.

More recently, several deep learning approaches have been proposed for MTSI. Among the others, deep autoregressive methods based on recurrent neural networks (RNNs) found widespread success (Lipton et al., 2016; Che et al., 2018; Luo et al., 2018; Yoon et al., 2018b; Cao et al., 2018). GRU-D (Che et al., 2018) learns how to process sequences with missing data by controlling the decay of the hidden states of a gated RNN. Cao et al. (2018) propose *BRITS*, a bidirectional GRU-D-like RNN for multivariate time series imputation that takes into account correlation among different channels to perform spatial imputation. Other successful strategies in the literature have been proposed that exploit the adversarial training framework to generate realistic reconstructed sequences (Yoon et al., 2018a; Fedus et al., 2018; Luo et al., 2018; 2019). Notably, GAIN (Yoon et al., 2018a) uses GANs (Goodfellow et al., 2014) to learn models that perform imputation in the i.i.d. settings. Luo et al. (2018; 2019) aim, instead, at learning models that generate realistic synthetic sequences and exploit them to fill missing values. Miao et al. (2021) use an approach similar to GAIN, but condition the generator on the predicted label for the target incomplete time series. Concurrently to our work, Kuppannagari et al. (2021) developed a graph-based spatio-temporal denoising autoencoder for spatio-temporal data coming from smart grids with known topology. Liu et al. (2019), instead, uses adversarial learning to train a multiscale model that imputes highly sparse time series in a hierarchical fashion. However, we argue that none of the above-cited methods can take full

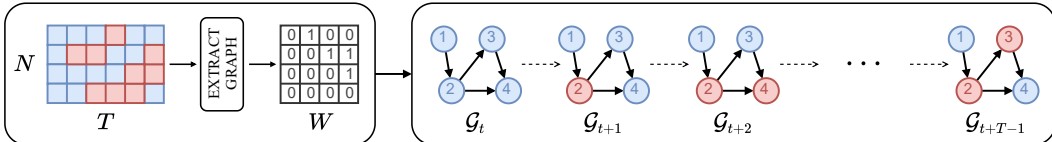

Figure 1: Representation of a multivariate time series as a sequence of graphs. Red circles denote nodes with missing values, nodes are identified.

advantage of relational information and nonlinear spatio-temporal dependencies. Most importantly, the above methods do not fully exploit the flexibility and expressiveness enabled by operating in the context of graph processing.

**Graph neural networks for TSA** Graph neural networks have been exploited in TSA mostly in spatio-temporal forecasting methods. The idea behind most of the methods present in the literature is to modify standard neural network architectures for sequential data by relying on operators that work in the graph domain. For example, Seo et al. (2018) propose a GRU cell where gates are implemented by spectral GNNs (Defferrard et al., 2016); Li et al. (2018) propose an analogous architecture replacing spectral GNNs with a diffusion-convolutional network (Atwood & Towsley, 2016). Note that these models are different w.r.t. approaches that use recurrent networks to propagate information graph-wise (Scarselli et al., 2008; Li et al., 2016). Yu et al. (2017) and Wu et al. (2019; 2020b) propose, instead, spatio-temporal convolutional neural networks that alternate convolutions on temporal and spatial dimensions. Similar approaches have also been studied in the context of attention-based models (Vaswani et al., 2017) with spatio-temporal Transformer-like architectures (Zhang et al., 2018; Cai et al., 2020). Another particularly interesting line of research is related to the problem of learning the graph structure underlying an input multivariate time series (Kipf et al., 2018; Wu et al., 2020b; Shang et al., 2020). While previously mentioned approaches focus on multivariate time series prediction, other methods aim at predicting changes in graph topology (Zambon et al., 2019; Paassen et al., 2020). Conversely, methods such as Temporal Graph Networks (Rossi et al., 2020) are tailored to learn node embeddings in dynamical graphs. Finally, recent works have proposed GNNs for imputing missing features in the context of i.i.d. data. Among the others, Spinelli et al. (2020) propose an adversarial framework to train GNNs on the data reconstruction task, while You et al. (2020) propose a bipartite graph representation for feature imputation. Lately, GNNs have also been exploited for spatial interpolation (Appleby et al., 2020; Wu et al., 2020a) – sometimes referred to as *kriging* (Stein, 1999). To the best of our knowledge, no previous GNN-based method targeted missing value imputation for generic multivariate time series.

## 3 PRELIMINARIES

**Sequences of graphs** We consider sequences of weighted directed graphs, where we observe a graph $\mathcal{G}_t$ with $N_t$ nodes at each time step $t$. A graph is a couple $\mathcal{G}_t = \langle \boldsymbol{X}_t, \boldsymbol{W}_t \rangle$, where $\boldsymbol{X}_t \in \mathbb{R}^{N_t \times d}$ is the node-attribute matrix whose $i$-th row contains the $d$-dimensional node-attribute vector $\boldsymbol{x}_t^i \in \mathbb{R}^d$ associated with the $i$-th node; entry $w_t^{i,j}$ of the adjacency matrix $\boldsymbol{W}_t \in \mathbb{R}^{N_t \times N_t}$ denotes the scalar weight of the edge (if any) connecting the $i$-th and $j$-th node. Fig. 1 exemplifies this modelling framework. We assume nodes to be identified, i.e., to have a unique ID that enables time-wise consistent processing. This problem setting can be easily extended to more general classes of graphs with attributed edges and global attributes. In this work, we mainly focus on problems where the topology of the graph is fixed and does not change over time, i.e., at each time step $\boldsymbol{W}_t = \boldsymbol{W}$ and $N_t = N$.

Any generic multivariate time series fits the above framework by letting each channel of the sequence (i.e., each sensor) correspond to a node and using the available relation information to build an adjacency matrix. If no relational information is available, one could use the identity matrix, but this would defeat the purpose of the formulation. A more proper choice of $\boldsymbol{W}_t$ can be made using any standard similarity score (e.g., Pearson correlation) or a (thresholded) kernel. A more advanced approach instead could aim at learning an adjacency directly from data by using, for instance, spatial attention scores or resorting to graph learning techniques, e.g., Kipf et al. (2018). From now on, we assume that input multivariate time series have homogeneous channels, i.e., sensors are of the same

type. Note that this assumption does not imply a loss in generality: it is always possible to standardize node features by adding *sensor type* attributes and additional dimensions to accommodate the different types of sensor readings. Alternatively, one might directly model the problem by exploiting heterogeneous graphs (Schlichtkrull et al., 2018).

**Multivariate time series imputation**   To model the presence of missing values, we consider, at each step, a binary mask $\boldsymbol{M}_t \in \{0, 1\}^{N_t \times d}$ where each row $\boldsymbol{m}_t^i$ indicates which of the corresponding node attributes of $\boldsymbol{x}_t^i$ are available in $\boldsymbol{X}_t$. It follows that, $m_t^{i,j} = 0$ implies $x_t^{i,j}$ to be missing; conversely, if $m_t^{i,j} = 1$, then $x_t^{i,j}$ stores the actual sensor reading. We denote by $\widetilde{\boldsymbol{X}_t}$ the *unknown* ground truth node-attribute matrix, i.e., the complete node-attribute matrix without any missing data. We assume stationarity of missing data distribution and, in experiments, we mostly focus on the missing at random (MAR) scenario (Rubin, 1976). We neither make assumptions on the number of concurrent sensor failures, nor on the length of missing data blocks, i.e., multiple failures extended over time are accounted for. Clearly, one should expect imputation performance to scale with the number of concurrent faults and the time length of missing data bursts.

The objective of MTSI is to impute missing values in a sequence of input data. More formally, given a graph sequence $\mathcal{G}_{[t,t+T]}$ of length $T$, we can define the missing data reconstruction error as

$$\mathcal{L}\left(\widehat{\boldsymbol{X}}_{[t,t+T]}, \widetilde{\boldsymbol{X}}_{[t,t+T]}, \overline{\boldsymbol{M}}_{[t,t+T]}\right) = \sum\nolimits_{h=t}^{t+T}\sum\nolimits_{i=1}^{N_t} \frac{\left\langle \overline{\boldsymbol{m}}_h^i, \ell\left(\hat{\boldsymbol{x}}_h^i, \tilde{\boldsymbol{x}}_h^i\right)\right\rangle}{\left\langle \overline{\boldsymbol{m}}_h^i, \overline{\boldsymbol{m}}_h^i\right\rangle}, \tag{1}$$

where $\hat{\boldsymbol{x}}_h^i$ is the reconstructed $\tilde{\boldsymbol{x}}_h^i$; $\overline{\boldsymbol{M}}_{[t,t+T]}$ and $\overline{\boldsymbol{m}}_h^i$ are respectively the logical binary complement of $\boldsymbol{M}_{[t,t+T]}$ and $\boldsymbol{m}_h^i$, $\ell(\,\cdot\,,\,\cdot\,)$ is an element-wise error function (e.g., absolute or squared error) and $\langle\,\cdot\,,\,\cdot\,\rangle$ indicates the standard dot product. Note that, in practice, it is impossible to have access to $\widetilde{\boldsymbol{X}}_{[t,t+T]}$ and, as a consequence, it is necessary to define a surrogate optimization objective by, for example, using a forecasting loss or generating synthetic missing values. In the context of trainable, parametric, imputation methods, we consider two different operational settings. In the first one, named *in-sample imputation*, the model is trained to reconstruct missing values in a given *fixed* input sequence $\boldsymbol{X}_{[t,t+T]}$, i.e., the model is trained on all the available data except those that are missing and those that have been removed from the sequence to emulate additional failures for evaluation. Differently, in the second one (referred to as *out-of-sample imputation*), the model is trained and evaluated on disjoint sequences. Note that in both cases the model does not have access to the ground-truth data used for the final evaluation. The first operational setting simulates the case where a practitioner fits the model directly on the sequence to fill up its gaps. The second, instead, simulates the case where one wishes to use a model fitted on a set of historical data to impute missing values in an unseen target sequence.

## 4   GRAPH RECURRENT IMPUTATION NETWORK

In this section, we present our approach, the *Graph Recurrent Imputation Network* (GRIN), a graph-based, recurrent neural architecture for MTSI. Given a multivariate time series $\boldsymbol{X}_{[t,t+T]}$ with mask $\boldsymbol{M}_{[t,t+T]}$, our objective is to reconstruct missing values in the input sequence by combining the information coming from both the temporal and spatial dimensions. To do so, we design a *novel bidirectional graph recurrent neural network* which progressively processes the input sequence both forward and backward in time by performing two stages of imputation for each direction. Then, a feed-forward network takes as input the representation learned by the forward and backward models and performs a final – refined – imputation for each node of the graph and step of the sequence. More precisely, the final imputation depends on the output of two GRIN modules whose learned representations are finally processed (space and time wise) by a last decoding multilayer perceptron (MLP). An overview of the complete architecture is given in Fig. 2. As shown in the figure, the two modules impute missing values iteratively, using at each time step previously imputed values as input. We proceed by first describing in detail the unidirectional model, and then we provide the bidirectional extension.

**Unidirectional model**   Each GRIN module is composed of two blocks, a spatio-temporal encoder and a spatial decoder, which process the input sequence of graphs in two stages. The spatio-temporal encoder maps the input sequence $\boldsymbol{X}_{[t,t+T]}$ to a spatio-temporal representation $\boldsymbol{H}_{[t,t+T]} \in \mathbb{R}^{N_t \times l}$ by

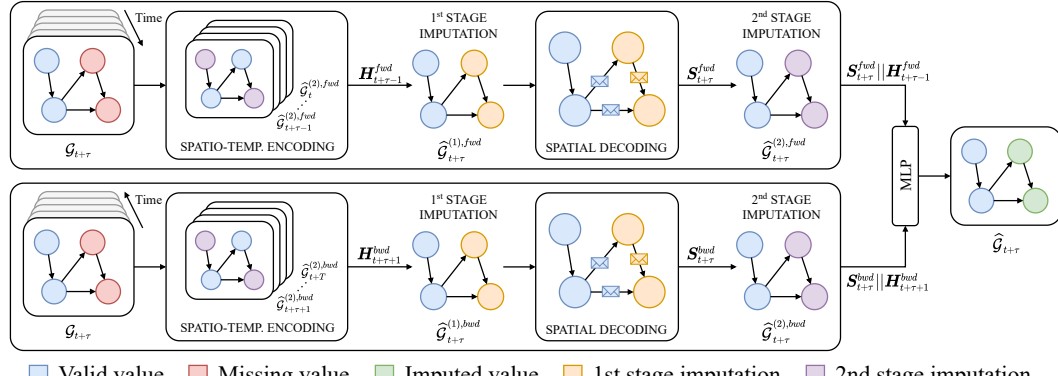

Figure 2: An overview of the bidirectional architecture. Here, each unidirectional GRIN module is processing the $\tau$-th step of an input sequence with 4 dimensions (sensors). Two values are missing at the considered time step. GRIN performs a first imputation, which is then processed and refined by the spatial decoder. These second-stage imputations are then used to continue the processing at the next step. An MLP processes learned representations node and time wise to obtain final imputations.

exploiting an ad-hoc designed recurrent GNN. The spatial decoder, instead, takes advantage of the learned representations to perform two consecutive rounds of imputation. A first-stage imputation is obtained from the representation by using a linear readout; the second one exploits available relational, spatial, information at time step $t$. In particular, the decoder is implemented by an MPNN which learns to infer the observed values at each $i$-th node – $\boldsymbol{x}_t^i$ – by refining first-stage imputations considering – locally – $\boldsymbol{H}_{t-1}$ and values observed at neighboring nodes.

**Spatio-temporal Encoder**   In the encoder, the input sequence $\boldsymbol{X}_{[t,t+T]}$ and mask $\boldsymbol{M}_{[t,t+T]}$ are processed sequentially one step at a time, by means of a recurrent neural network with gates implemented by message-passing layers. Any message-passing operator could be used in principle. In particular, given $\boldsymbol{z}_{t,k-1}^i$, i.e., the node features vector at layer $k-1$, we consider the general class of MPNNs described as

$$\text{MPNN}_k\big(\boldsymbol{z}_{t,k-1}^i, \boldsymbol{W}_t\big) = \gamma_k\Big(\boldsymbol{z}_{t,k-1}^i, \textstyle\sum_{j \in \mathcal{N}(i)} \rho_k\big(\boldsymbol{z}_{t,k-1}^i, \boldsymbol{z}_{t,k-1}^j\big)\Big) = \boldsymbol{z}_{t,k}^i, \tag{2}$$

where $\mathcal{N}(i)$ is the set of neighbors of the $i$-th node in $\mathcal{G}_t$, $\gamma_k$ and $\rho_k$ are generic, differentiable, update and message functions (e.g., MLPs), and $\Sigma$ is a permutation invariant, differentiable aggregation function (e.g., $sum$ or $mean$). Note that several definitions of *neighborhood* are possible, e.g., one might consider nodes connected by paths up to a certain length $l$. For the sake of simplicity, from now on, we indicate with $\text{MPNN}(\boldsymbol{z}_t^i, \boldsymbol{W}_t)$ the forward pass of a generic $K$-layered message-passing neural network. In the following, we use MPNNs as the building blocks for our spatio-temporal feature extractors. To learn the dynamics of the system, we leverage on gated recurrent units (GRUs; Cho et al., 2014). As previously mentioned, similarly to Seo et al. (2018) and Li et al. (2018), we implement the GRU gates by relying on the message-passing layers defined above. At the node level, the elements of the message-passing GRU (MPGRU) can be described as:

$$\boldsymbol{r}_t^i = \sigma\big(\text{MPNN}\big(\big[\hat{\boldsymbol{x}}_t^{i(2)}||\boldsymbol{m}_t^i||\boldsymbol{h}_{t-1}^i\big], \boldsymbol{W}_t\big)\big) \tag{3}$$

$$\boldsymbol{u}_t^i = \sigma\big(\text{MPNN}\big(\big[\hat{\boldsymbol{x}}_t^{i(2)}||\boldsymbol{m}_t^i||\boldsymbol{h}_{t-1}^i\big], \boldsymbol{W}_t\big)\big) \tag{4}$$

$$\boldsymbol{c}_t^i = \tanh\big(\text{MPNN}\big(\big[\hat{\boldsymbol{x}}_t^{i(2)}||\boldsymbol{m}_t^i||\boldsymbol{r}_t^i \odot \boldsymbol{h}_{t-1}^i\big], \boldsymbol{W}_t\big)\big) \tag{5}$$

$$\boldsymbol{h}_t^i = \boldsymbol{u}_t^i \odot \boldsymbol{h}_{t-1}^i + (1 - \boldsymbol{u}_t^i) \odot \boldsymbol{c}_t^i \tag{6}$$

where $\boldsymbol{r}_t^i, \boldsymbol{u}_t^i$ are the reset and update gates, respectively, $\boldsymbol{h}_t^i$ is the hidden representation of the $i$-th node at time $t$, and $\hat{\boldsymbol{x}}_t^{i(2)}$ is the output of the decoding block at the previous time-step (see next paragraph). The symbols $\odot$ and $||$ denote the Hadamard product and the concatenation operator, respectively. The initial representation $\boldsymbol{H}_{t-1}$ can either be initialized as a constant or with a learnable embedding. Note that for the steps where input data are missing, the encoder is fed with predictions from the decoder block, as explained in the next subsection. By carrying out the above computation time and node wise, we get the encoded sequence $\boldsymbol{H}_{[t,t+T]}$.

**Spatial Decoder**   As a first decoding step, we generate one-step-ahead predictions from the hidden representations of the MPGRU by means of a linear readout

$$\widehat{\boldsymbol{Y}}_t^{(1)} = \boldsymbol{H}_{t-1}\boldsymbol{V}_h + \boldsymbol{b}_h, \tag{7}$$

where $\boldsymbol{V}_h \in \mathbb{R}^{l \times d}$ is a learnable weight matrix and $\boldsymbol{b}_h \in \mathbb{R}^d$ is a learnable bias vector. We then define the *filler* operator as

$$\Phi(\boldsymbol{Y}_t) = \boldsymbol{M}_t \odot \boldsymbol{X}_t + \overline{\boldsymbol{M}}_t \odot \boldsymbol{Y}_t; \tag{8}$$

intuitively, the filler operator replaces the missing values in the input $\boldsymbol{X}_t$ with the values at the same positions in $\boldsymbol{Y}_t$. By feeding $\widehat{\boldsymbol{Y}}_t^{(1)}$ to the filler operator, we get the *first-stage imputation* $\widehat{\boldsymbol{X}}_t^{(1)}$ such that the output is $\boldsymbol{X}_t$ with missing values replaced by the one-step-ahead predictions $\widehat{\boldsymbol{Y}}_t^{(1)}$. The resulting node-level predictions are then concatenated to the mask $\boldsymbol{M}_t$ and the hidden representation $\boldsymbol{H}_{t-1}$, and processed by a final *one-layer* MPNN which computes for each node an *imputation representation* $\boldsymbol{s}_t^i$ as

$$\boldsymbol{s}_t^i = \gamma\Big(\boldsymbol{h}_{t-1}^i, \sum\nolimits_{j \in \mathcal{N}(i)/i} \rho\big([\Phi(\hat{\boldsymbol{x}}_t^{j(1)})||\boldsymbol{h}_{t-1}^j||\boldsymbol{m}_t^j]\big)\Big). \tag{9}$$

Notice that, as previously highlighted, the imputation representations only depend on messages received from neighboring nodes and the representation at the previous step. In fact, by aggregating only messages from the one-hop neighborhood, the representations $\boldsymbol{s}_t^i$ are independent of the input features $\boldsymbol{x}_t^i$ of the $i$-th node itself. This constraint forces the model to learn how to reconstruct a target input by taking into account spatial dependencies: this has a regularizing effect since the model is constrained to focus on *local* information. Afterward, we concatenate imputation representation $\boldsymbol{S}_t$ with hidden representation $\boldsymbol{H}_{t-1}$, and generate second-stage imputations by using a second linear readout and applying the filler operator:

$$\widehat{\boldsymbol{Y}}_t^{(2)} = [\boldsymbol{S}_t||\boldsymbol{H}_{t-1}]\,\boldsymbol{V}_s + \boldsymbol{b}_s; \qquad \widehat{\boldsymbol{X}}_t^{(2)} = \Phi(\widehat{\boldsymbol{Y}}_t^{(2)}) \tag{10}$$

Finally, we feed $\widehat{\boldsymbol{X}}_t^{(2)}$ as input to the MPGRU (Eq. 3 – 6) to update the hidden representation and proceed to process the next input graph $\mathcal{G}_{t+1}$.

**Bidirectional Model**   Extending GRIN to account for both forward and backward dynamics is straightforward and can be achieved by duplicating the architecture described in the two previous paragraphs. The first module will process the sequence in the forward direction (from the beginning of the sequence towards its end), while the second one in the other way around. The final imputation is then obtained with an MLP aggregating representations extracted by the two modules:

$$\widehat{\boldsymbol{y}}_t^i = \text{MLP}\big([\boldsymbol{s}_t^{i,fwd}||\boldsymbol{h}_{t-1}^{i,fwd}||\boldsymbol{s}_t^{i,bwd}||\boldsymbol{h}_{t+1}^{i,bwd}]\big), \tag{11}$$

where $fwd$ and $bwd$ denote the forward and backward modules, respectively. The final output can then be easily obtained as $\widehat{\boldsymbol{X}}_{[t,t+T]} = \Phi(\widehat{\boldsymbol{Y}}_{[t,t+T]})$. Note that, by construction, our model can exploit all the available relevant spatio-temporal information, since the only value explicitly masked out for each node is $\boldsymbol{x}_t^i$. At the same time, it is important to realize that our model does not merely reconstruct the input as an autoencoder, but it is specifically tailored for the imputation task due to its inductive biases. The model is trained by minimizing the reconstruction error of all imputation stages in both directions (see Appendix A).

## 5   EMPIRICAL EVALUATION

In this section, we empirically evaluate our approach against state-of-the-art baselines on four datasets coming from three relevant application domains. Our approach, remarkably, achieves state-of-the-art performance on all of them.

- **Air Quality (AQI)**: dataset of recordings of several air quality indices from 437 monitoring stations spread across 43 Chinese cities. We consider only the PM2.5 pollutant. Prior works on imputation (Yi et al., 2016; Cao et al., 2018) consider a reduced version of this dataset, including only 36 sensors (**AQI-36** in the following). We evaluate our model on both datasets. We use as adjacency matrix a thresholded Gaussian kernel (Shuman et al., 2013) computed from pairwise geographic distances.

Table 1: Results on the air datasets. Performance averaged over 5 runs.

| D | M | In-sample | | | Out-of-sample | | |
|---|---|---|---|---|---|---|---|
| | | MAE | MSE | MRE (%) | MAE | MSE | MRE (%) |
| AQI-36 | Mean | $53.48_{\pm 0.00}$ | $4578.08_{\pm 00.00}$ | $76.77_{\pm 0.00}$ | $53.48_{\pm 0.00}$ | $4578.08_{\pm 00.00}$ | $76.77_{\pm 0.00}$ |
| | KNN | $30.21_{\pm 0.00}$ | $2892.31_{\pm 00.00}$ | $43.36_{\pm 0.00}$ | $30.21_{\pm 0.00}$ | $2892.31_{\pm 00.00}$ | $43.36_{\pm 0.00}$ |
| | MF | $30.54_{\pm 0.26}$ | $2763.06_{\pm 63.35}$ | $43.84_{\pm 0.38}$ | – | – | – |
| | MICE | $29.89_{\pm 0.11}$ | $2575.53_{\pm 07.67}$ | $42.90_{\pm 0.15}$ | $30.37_{\pm 0.09}$ | $2594.06_{\pm 07.17}$ | $43.59_{\pm 0.13}$ |
| | VAR | $13.16_{\pm 0.21}$ | $513.90_{\pm 12.39}$ | $18.89_{\pm 0.31}$ | $15.64_{\pm 0.08}$ | $833.46_{\pm 13.85}$ | $22.02_{\pm 0.11}$ |
| | rGAIN | $12.23_{\pm 0.17}$ | $393.76_{\pm 12.66}$ | $17.55_{\pm 0.25}$ | $15.37_{\pm 0.26}$ | $641.92_{\pm 33.89}$ | $21.63_{\pm 0.36}$ |
| | BRITS | $12.24_{\pm 0.26}$ | $495.94_{\pm 43.56}$ | $17.57_{\pm 0.38}$ | $14.50_{\pm 0.35}$ | $662.36_{\pm 65.16}$ | $20.41_{\pm 0.50}$ |
| | MPGRU | $12.46_{\pm 0.35}$ | $517.21_{\pm 41.02}$ | $17.88_{\pm 0.50}$ | $16.79_{\pm 0.52}$ | $1103.04_{\pm 106.83}$ | $23.63_{\pm 0.73}$ |
| | **GRIN** | $\mathbf{10.51}_{\pm 0.28}$ | $\mathbf{371.47}_{\pm 17.38}$ | $\mathbf{15.09}_{\pm 0.40}$ | $\mathbf{12.08}_{\pm 0.47}$ | $\mathbf{523.14}_{\pm 57.17}$ | $\mathbf{17.00}_{\pm 0.67}$ |
| AQI | Mean | $39.60_{\pm 0.00}$ | $3231.04_{\pm 00.00}$ | $59.25_{\pm 0.00}$ | $39.60_{\pm 0.00}$ | $3231.04_{\pm 00.00}$ | $59.25_{\pm 0.00}$ |
| | KNN | $34.10_{\pm 0.00}$ | $3471.14_{\pm 00.00}$ | $51.02_{\pm 0.00}$ | $34.10_{\pm 0.00}$ | $3471.14_{\pm 00.00}$ | $51.02_{\pm 0.00}$ |
| | MF | $26.74_{\pm 0.24}$ | $2021.44_{\pm 27.98}$ | $40.01_{\pm 0.35}$ | – | – | – |
| | MICE | $26.39_{\pm 0.13}$ | $1872.53_{\pm 15.97}$ | $39.49_{\pm 0.19}$ | $26.98_{\pm 0.10}$ | $1930.92_{\pm 10.08}$ | $40.37_{\pm 0.15}$ |
| | VAR | $18.13_{\pm 0.84}$ | $918.68_{\pm 56.55}$ | $27.13_{\pm 1.26}$ | $22.95_{\pm 0.30}$ | $1402.84_{\pm 52.63}$ | $33.99_{\pm 0.44}$ |
| | rGAIN | $17.69_{\pm 0.17}$ | $861.66_{\pm 17.49}$ | $26.48_{\pm 0.25}$ | $21.78_{\pm 0.50}$ | $1274.93_{\pm 60.28}$ | $32.26_{\pm 0.75}$ |
| | BRITS | $17.24_{\pm 0.13}$ | $924.34_{\pm 18.26}$ | $25.79_{\pm 0.20}$ | $20.21_{\pm 0.22}$ | $1157.89_{\pm 25.66}$ | $29.94_{\pm 0.33}$ |
| | MPGRU | $15.80_{\pm 0.05}$ | $816.39_{\pm 05.99}$ | $23.63_{\pm 0.08}$ | $18.76_{\pm 0.11}$ | $1194.35_{\pm 15.23}$ | $27.79_{\pm 0.16}$ |
| | **GRIN** | $\mathbf{13.10}_{\pm 0.08}$ | $\mathbf{615.80}_{\pm 10.09}$ | $\mathbf{19.60}_{\pm 0.11}$ | $\mathbf{14.73}_{\pm 0.15}$ | $\mathbf{775.91}_{\pm 28.49}$ | $\mathbf{21.82}_{\pm 0.23}$ |

- **Traffic**: we consider the **PEMS-BAY** and **METR-LA** datasets from Li et al. (2018), containing data from traffic sensors from the San Francisco Bay Area and the Los Angeles County Highway. We use the same approach of Li et al. (2018) and Wu et al. (2019) to obtain an adjacency matrix.

- **Smart grids**: we consider data from the Irish Commission for Energy Regulation Smart Metering Project (**CER-E**; Commission for Energy Regulation, 2016). We select only the subset of the available smart meters monitoring the energy consumption of small and medium-sized enterprises (SMEs), i.e., $485$ time series with samples acquired every 30 minutes. We build an adjacency matrix by extracting a k-nearest neighbor graph (with $k = 10$) from the similarity matrix built by computing the correntropy (Liu et al., 2007) among the time series.

For the air quality datasets, we adopt the same evaluation protocol of previous works (Yi et al., 2016; Cao et al., 2018) and we show results for both the in-sample and out-of-sample settings. For the traffic and energy consumption datasets, we consider only the out-of-sample scenario (except for matrix factorization which only works in-sample). We simulate the presence of missing data by considering 2 different settings: 1) *Block missing*, i.e, at each step, for each sensor, we randomly drop $5\%$ of the available data and, in addition, we simulate a failure with probability $p_{failure} = 0.15\%$ and sample its duration uniformly in the interval $[min\_steps, max\_steps]$, where $min\_steps$ and $max\_steps$ are the number of time steps corresponding respectively to 1 and 4 hours in the traffic case and 2 hours and 2 days for CER-E; 2) *Point missing*, i.e., we simply randomly mask out $25\%$ of the available data. We split all the datasets into training/validation/test sets. We use as performance metrics the *mean absolute error* (MAE), *mean squared error* (MSE) and *mean relative error* (MRE; Cao et al., 2018) computed over the imputation window. For all the experiments, we use as message-passing operator the diffusion convolution introduced by Atwood & Towsley (2016). We consider BRITS (Cao et al., 2018) as the principal competing alternative among non-adversarial deep autoregressive approaches, as it shares architectural similarities with our methods. As additional baselines we consider: 1) MEAN, i.e., imputation using the node-level average; 2) KNN, i.e., imputation by averaging values of the $k = 10$ neighboring nodes with the highest weight in the adjacency matrix $\boldsymbol{W}_t$; 3) MICE (White et al., 2011), limiting the maximum number of iterations to 100 and the number of nearest features to 10; 4) Matrix Factorization (MF) with $rank = 10$; 5) VAR, i.e., a *Vector Autoregressive* one-step-ahead predictor; 6) rGAIN, i.e., an unsupervised version of SSGAN (Miao et al., 2021) which can be seen as GAIN (Yoon et al., 2018a) with bidirectional recurrent encoder and decoder; 7) MPGRU, a one-step-ahead GNN-based predictor similar to DCRNN (Li et al., 2018).

Table 2: Results on the traffic and smart grids datasets. Performance averaged over 5 runs.

| D | M | Block missing | | | Point missing | | |
|---|---|---|---|---|---|---|---|
| | | MAE | MSE | MRE(%) | MAE | MSE | MRE(%) |
| PEMS-BAY | Mean | 5.46 ± 0.00 | 87.56 ± 0.00 | 8.75 ± 0.00 | 5.42 ± 0.00 | 86.59 ± 0.00 | 8.67 ± 0.00 |
| | KNN | 4.30 ± 0.00 | 49.90 ± 0.00 | 6.90 ± 0.00 | 4.30 ± 0.00 | 49.80 ± 0.00 | 6.88 ± 0.00 |
| | MF | 3.28 ± 0.01 | 50.14 ± 0.13 | 5.26 ± 0.01 | 3.29 ± 0.01 | 51.39 ± 0.64 | 5.27 ± 0.02 |
| | MICE | 2.94 ± 0.02 | 28.28 ± 0.37 | 4.71 ± 0.03 | 3.09 ± 0.02 | 31.43 ± 0.41 | 4.95 ± 0.02 |
| | VAR | 2.09 ± 0.10 | 16.06 ± 0.73 | 3.35 ± 0.16 | 1.30 ± 0.00 | 6.52 ± 0.01 | 2.07 ± 0.01 |
| | rGAIN | 2.18 ± 0.01 | 13.96 ± 0.20 | 3.50 ± 0.02 | 1.88 ± 0.02 | 10.37 ± 0.20 | 3.01 ± 0.04 |
| | BRITS | 1.70 ± 0.01 | 10.50 ± 0.07 | 2.72 ± 0.01 | 1.47 ± 0.00 | 7.94 ± 0.03 | 2.36 ± 0.00 |
| | MPGRU | 1.59 ± 0.00 | 14.19 ± 0.11 | 2.56 ± 0.01 | 1.11 ± 0.00 | 7.59 ± 0.02 | 1.77 ± 0.00 |
| | **GRIN** | **1.14** ± **0.01** | **6.60** ± **0.10** | **1.83** ± **0.02** | **0.67** ± **0.00** | **1.55** ± **0.01** | **1.08** ± **0.00** |
| METR-LA | Mean | 7.48 ± 0.00 | 139.54 ± 0.00 | 12.96 ± 0.00 | 7.56 ± 0.00 | 142.22 ± 0.00 | 13.10 ± 0.00 |
| | KNN | 7.79 ± 0.00 | 124.61 ± 0.00 | 13.49 ± 0.00 | 7.88 ± 0.00 | 129.29 ± 0.00 | 13.65 ± 0.00 |
| | MF | 5.46 ± 0.02 | 109.61 ± 0.78 | 9.46 ± 0.04 | 5.56 ± 0.03 | 113.46 ± 1.08 | 9.62 ± 0.05 |
| | MICE | 4.22 ± 0.05 | 51.07 ± 1.25 | 7.31 ± 0.09 | 4.42 ± 0.07 | 55.07 ± 1.46 | 7.65 ± 0.12 |
| | VAR | 3.11 ± 0.08 | 28.00 ± 0.76 | 5.38 ± 0.13 | 2.69 ± 0.00 | 21.10 ± 0.02 | 4.66 ± 0.00 |
| | rGAIN | 2.90 ± 0.01 | 21.67 ± 0.15 | 5.02 ± 0.02 | 2.83 ± 0.01 | 20.03 ± 0.09 | 4.91 ± 0.01 |
| | BRITS | 2.34 ± 0.01 | 17.00 ± 0.14 | 4.05 ± 0.01 | 2.34 ± 0.00 | 16.46 ± 0.05 | 4.05 ± 0.00 |
| | MPGRU | 2.57 ± 0.01 | 25.15 ± 0.17 | 4.44 ± 0.01 | 2.44 ± 0.00 | 22.17 ± 0.03 | 4.22 ± 0.00 |
| | **GRIN** | **2.03** ± **0.00** | **13.26** ± **0.05** | **3.52** ± **0.01** | **1.91** ± **0.00** | **10.41** ± **0.03** | **3.30** ± **0.00** |
| CER-E | Mean | 1.49 ± 0.00 | 5.96 ± 0.00 | 72.47 ± 0.00 | 1.51 ± 0.00 | 6.09 ± 0.00 | 71.51 ± 0.00 |
| | KNN | 1.15 ± 0.00 | 6.53 ± 0.00 | 56.11 ± 0.00 | 1.22 ± 0.00 | 7.23 ± 0.00 | 57.71 ± 0.00 |
| | MF | 0.97 ± 0.01 | 4.38 ± 0.06 | 47.20 ± 0.31 | 1.01 ± 0.01 | 4.65 ± 0.07 | 47.87 ± 0.36 |
| | MICE | 0.96 ± 0.01 | 3.08 ± 0.03 | 46.65 ± 0.44 | 0.98 ± 0.00 | 3.21 ± 0.04 | 46.59 ± 0.23 |
| | VAR | 0.64 ± 0.03 | 1.75 ± 0.06 | 31.21 ± 1.60 | 0.53 ± 0.00 | 1.26 ± 0.00 | 24.94 ± 0.02 |
| | rGAIN | 0.74 ± 0.00 | 1.77 ± 0.02 | 36.06 ± 0.14 | 0.71 ± 0.00 | 1.62 ± 0.02 | 33.45 ± 0.16 |
| | BRITS | 0.64 ± 0.00 | 1.61 ± 0.01 | 31.05 ± 0.05 | 0.64 ± 0.00 | 1.59 ± 0.01 | 30.07 ± 0.11 |
| | MPGRU | 0.53 ± 0.00 | 1.84 ± 0.01 | 25.88 ± 0.09 | 0.41 ± 0.00 | 1.22 ± 0.01 | 19.51 ± 0.03 |
| | **GRIN** | **0.42** ± **0.00** | **1.07** ± **0.01** | **20.24** ± **0.04** | **0.29** ± **0.00** | **0.53** ± **0.00** | **13.71** ± **0.03** |

We provide further comment and in depth details on baselines and datasets, together with additional experiments on synthetic data in the appendix.

## 5.1 RESULTS

Empirical results show that GRIN can achieve large improvements in imputation performance on several scenarios, as well as increased flexibility. In fact, differently from the other state-of-the-art baselines, GRIN can handle input with a variable number of dimensions. Tab. 1 shows the experimental results on the air quality datasets. In the in-sample settings, we compute metrics using as imputation the value obtained by averaging predictions over all the overlapping windows; in the out-of-sample settings, instead, we simply report results by averaging the *error* over windows. GRIN largely outperforms other baselines on both settings. In particular, in the latter case, GRIN decreases MAE w.r.t. the closest baseline by more than 20% in AQI. Interestingly, GRIN consistently outperforms BRITS in imputing missing values also for sensors corresponding to isolated (disconnected) nodes, i.e., nodes corresponding to stations more than 40 km away from any other station (see B.1): this is empirical evidence of the positive regularizations encoded into GRIN. Our method achieves more accurate imputation also in the 36-dimensional dataset, where we could expect the graph representation to have a lower impact. Results for the traffic and smart grids datasets are shown in Tab. 2. In the traffic dataset, our method outperforms both BRITS and rGAIN by a wide margin in all the considered settings while using a much lower number of parameters (see A). In the traffic datasets, on average, GRIN reduces MAE by ≈ 29% w.r.t. BRITS and, in particular, in the Point missing setting of the PEMS-BAY dataset, the error is halved. In CER-E, GRIN consistently outperforms other baselines. Besides show-

ing the effectiveness of our approach in a relevant application field, this experiment also goes to show that GRIN can be exploited in settings where relational information is not readily available.

Finally, Tab. 3 show results – in terms of MAE – of an ablation study on the out-of-sample scenario in AQI, METR-LA (in the Block Missing settings), and CER-E (Point Missing setting). In particular, we compare GRIN against 3 baselines to assess the impact of the spatial decoder and of the bidirectional architecture. The first baseline is essentially a bidirectional MPGRU

Table 3: Ablation study. Averages over 5 runs.

| Model | AQI | METR-LA | CER-E |
|---|---|---|---|
| **GRIN** | **14.73** $\pm$ **0.15** | **2.03** $\pm$ **0.00** | **0.29** $\pm$ **0.00** |
| w/o sp. dec. | 15.40 $\pm$ 0.14 | 2.32 $\pm$ 0.01 | **0.29** $\pm$ **0.00** |
| w/ denoise dec. | 17.23 $\pm$ 1.12 | 2.96 $\pm$ 0.18 | 0.32 $\pm$ 0.00 |
| MPGRU | 18.76 $\pm$ 0.11 | 2.57 $\pm$ 0.01 | 0.41 $\pm$ 0.00 |

where values are imputed by a final MLP taking as inputs $\mathbf{h}_{t-1}^{fwd}$ and $\mathbf{h}_{t+1}^{bwd}$, while the second one has an analogous architecture, but uses hidden representation and time step $t$ (for both directions) and, thus, behaves similarly to a denoising autoencoder. As reference, we report the results of the unidirectional MPGRU. Results show that the components we introduce do contribute to significantly reduce imputation error. It is clear that spatial decoding and the bidirectional architecture are important to obtain accurate missing data reconstruction, especially in realistic settings with blocks of missing data. Interestingly, the denoising model suffers in the Block missing scenario, while, as one might expect, works well in the Point Missing setting. For additional results and discussion about scalability issues, we refer to the appendix of the paper.

## 5.2 VIRTUAL SENSING

As a final experiment, we provide a quantitative and qualitative assessment of the proposed method in virtual sensing. The idea (often studied in the context of *kriging* – see Section 2) is to simulate the presence of a sensor by adding a node with no available data and, then, let the model reconstruct the corresponding time series. Note that for the approach to work several assumptions are needed: 1) we have to assume that the physical quantity being monitored can be reconstructed from observations at neighboring sensors; 2) we should assume a high-degree of homogeneity of sensors (e.g., in the case of air quality stations we should assume that sensors are placed at the same height) or that the features characterizing each neighboring sen-

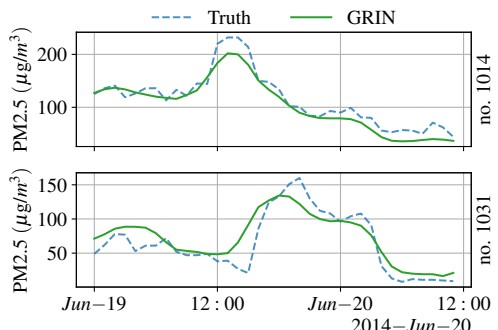

Figure 3: Reconstruction of observations from sensors removed from the training set. Plots show that GRIN might be used for virtual sensing.

sor (e.g., placement) are available to the model. In this context, it is worth noting that, due to the inductive biases embedded in the model, GRIN performs reconstruction not only by minimizing reconstruction error at the single node, but by regularizing the reconstructed value for imputation at neighboring sensors. We masked out observed values of the two nodes of AQI-36 with highest (station no. 1014) and lowest (no. 1031) connectivity, and trained GRIN on the remaining part of the data as usual. Results, in Fig. 3, qualitatively show that GRIN can infer the trend and scale for unseen sensors. In terms of MAE, GRIN scored 11.74 for sensor 1014 and 20.00 for sensor 1031 (averages over 5 independent runs).

## 6 CONCLUSIONS

We presented GRIN, a novel approach for MTSI exploiting modern graph neural networks. Our method imputes missing data by leveraging the relational information characterizing the underlying network of sensors and the functional dependencies among them. Compared against state-of-the-art baselines, our framework offers higher flexibility and achieves better reconstruction accuracy on all the considered scenarios. There are several possible directions for future works. From a theoretical perspective, it would be interesting to study the properties that would guarantee an accurate reconstruction. Furthermore, future work should study extensions able to deal with a non-stationary setting and further assess applications of GRIN in virtual and active sensing.

## REPRODUCIBILITY STATEMENT

Code to reproduce experiments presented in the paper is provided as supplementary material together with configuration files to replicate reported results. All datasets, except CER-E, are open and downloading links are provided in the supplementary material. The CER-E dataset can be obtained free of charge for research purposes (see appendix). For experiments where failures are simulated, we use random number generators with fixed seed for missing data generation to ensure reproducibility and consistency among experiments and baselines.

## ACKNOWLEDGMENTS

This research is funded by the Swiss National Science Foundation project 200021_172671: "ALPS-FORT: A Learning graPh-baSed framework FOr cybeR-physical sysTems". The authors wish to thank the Institute of Computational Science at USI for granting access to computational resources.

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

## APPENDIX

## A   DETAILED EXPERIMENTAL SETTINGS

In this appendix, we give more details on the experimental settings used to evaluate our approach. We train all the models by sampling at random 160 batches of 32 elements for each epoch, we fix the maximum number of epochs to 300 and we use early stopping on the validation set with a patience of 40 epochs. All methods are trained using a cosine learning rate scheduler with initial value of 0.001, decayed over the 300 training epochs. During training, we randomly mask out an additional 5% of the input data for each batch to foster robustness to noise and missing data.

For GRIN we minimize the following loss function is

$$
\begin{aligned}
\mathcal{L} = {} & \mathcal{L}\left(\boldsymbol{Y}_{[t,t+T]}, \boldsymbol{X}_{[t,t+T]}, \boldsymbol{M}_{[t,t+T]}\right) \\
& + \mathcal{L}\left(\boldsymbol{Y}^{(1),fwd}_{[t,t+T]}, \boldsymbol{X}_{[t,t+T]}, \boldsymbol{M}_{[t,t+T]}\right) + \mathcal{L}\left(\boldsymbol{Y}^{(2),fwd}_{[t,t+T]}, \boldsymbol{X}_{[t,t+T]}, \boldsymbol{M}_{[t,t+T]}\right) \\
& + \mathcal{L}\left(\boldsymbol{Y}^{(1),bwd}_{[t,t+T]}, \boldsymbol{X}_{[t,t+T]}, \boldsymbol{M}_{[t,t+T]}\right) + \mathcal{L}\left(\boldsymbol{Y}^{(2),bwd}_{[t,t+T]}, \boldsymbol{X}_{[t,t+T]}, \boldsymbol{M}_{[t,t+T]}\right),
\end{aligned}
$$

where each $\mathcal{L}\left(\,\cdot\,,\,\cdot\,,\,\cdot\,\right)$ is of the form of Eq. 1 and the element-wise error function is MAE. Note that here we are using $\boldsymbol{X}_{[t,t+T]}$ and $\boldsymbol{M}_{[t,t+T]}$ instead of $\widehat{\boldsymbol{X}}_{[t,t+T]}$ and $\overline{\boldsymbol{M}}_{[t,t+T]}$.

For BRITS, we use the same network hyperparameters of Cao et al. (2018) for the AQI-36 dataset. To account for the larger input dimension, for the other datasets we increase the number of hidden neurons in the RNNs cells to 128 for AQI/METR-LA and 256 for PEMS-BAY/CER-E. The number of neurons was tuned on the validation sets. For rGAIN we use the same number of units in the cells of the bidirectional RNN used by BRITS, but we concatenate a random vector (sampled from a uniform distribution) of dimension $z = 4$ to the input vector in order to model the sampling of the data generating process. To obtain predictions, we average out the outputs of $k = 5$ forward passes. For VAR we used an order of 5 and trained the model with SGD. Since the VAR model needs past 5 observations to predict the next step, we pad each sequence using the mean for each channel. Here we used a batch size to 64 and a learning rate of 0.0005. The order was selected with a small search in the range $[2, 12]$: we found out a window size of 5 to be ideal for all the considered datasets. For GRIN we use the same hyperparameters in all the datasets: a hidden dimension of 64 neurons for both the spatio-temporal encoder and the spatial decoder and of 64 neurons for the MLP. We use diffusion convolution as message-passing operation, with a diffusion step $k = 2$ in the spatio-temporal encoder and $k = 1$ in the temporal decoder. Note that, due to the architectural differences, the other neural network baselines have a number of parameters that is far higher than GRIN (depending on the considered dataset, up to $\approx 4M$ against $\approx 200K$). For MPGRU we use the same hyperparameters of GRIN (64 units for both the spatio-temporal encoder and the decoder).

For data processing we use the same steps of Li et al. (2018), data are normalized across the feature dimension (which means graph-wise for GRIN and node-wise for BRITS/rGAIN/VAR). Data masked out for evaluation are never used to train any model.

All the models were developed in Python (Van Rossum & Drake, 2009) using the following open-source libraries:

- PyTorch (Paszke et al., 2019);

- numpy (Harris et al., 2020);

- Neptune[1] (neptune.ai, 2021);

- scikit-learn (Pedregosa et al., 2011);

- fancyimpute (Rubinsteyn & Feldman).

---

[1] https://neptune.ai/

The implementation of the diffusion convolutional operator was adapted from the Graph-WaveNet codebase [2]. For the implementation of BRITS, we used the code provided by the authors[3]. The code to reproduce the experiments of the paper is available online[4].

Table 4: Statistics on adjacency matrices used in the experiments. Self loops are excluded.

|  | GRAPH | | | N. NEIGHBORS | | |
|---|---|---|---|---|---|---|
| Dataset | type | nodes | edges | mean | median | isolated nodes |
| AQI | undirected | 437 | 2699 | 12.35 | 9.0 | 14 |
| CER-E | directed | 485 | 4365 | 9.0 | 9.0 | 0 |
| PEMS-BAY | directed | 325 | 2369 | 7.29 | 7.0 | 12 |
| METR-LA | directed | 207 | 1515 | 7.32 | 7.0 | 5 |

## B DATASETS

In this appendix, we provide more details on the datasets that we used to run experiments. Tab. 4 shows detailed statistics for graph structure associated with each dataset, while Fig. 4 shows the corresponding adjacency matrices. Tab. 5 shows missing data statistics. In the following subsections, we go deeper into details for each dataset.

Table 5: Statistics on missing data distribution. (P) and (B) indicate the Point Missing and Block Missing settings, respectively. With *block*, we refer to missing data bursts longer than 2 time steps and shorter than or equal to 48.

|  |  | ORIGINAL DATA | | | INJECTED FAULTS | | |
|---|---|---|---|---|---|---|---|
| D |  | % missing | avg. block | median block | % | avg. block | median block |
| AQI |  | 25.67 | 6.69 | 4.0 | 10.67 | 7.59 | 4.0 |
| AQI-36 |  | 13.24 | 7.24 | 4.0 | 11.33 | 6.52 | 4.0 |
| PEMS-BAY | (P) | 0.02 | 12.0 | 12.0 | 25.0 | 3.33 | 3.0 |
|  | (B) |  |  |  | 9.07 | 27.26 | 28.0 |
| METR-LA | (P) | 8.10 | 12.44 | 9.0 | 23.00 | 3.33 | 3.0 |
|  | (B) |  |  |  | 8.4 | 25.68 | 26.0 |
| CER-E | (P) | 0.04 | 48.0 | 48.0 | 24.97 | 3.33 | 3.0 |
|  | (B) |  |  |  | 8.38 | 22.45 | 21.0 |

### B.1 AIR QUALITY

Air pollution is nowadays a ubiquitous problem. The Urban Computing project (Zheng et al., 2014; 2015) published several datasets containing real measurements of different indices affecting human life in urban spaces. We consider as benchmark the dataset regarding the air quality index (*AQI*). The complete dataset contains hourly measurements of six pollutants from 437 air quality monitoring stations, spread over 43 cities in China, over a period of one year (from May 2014 to April 2015). Prior works on imputation (Yi et al., 2016; Cao et al., 2018) considered a reduced version of this dataset, including only 36 sensors (*AQI-36*). This dataset is particularly interesting as a benchmark for imputation due to the high rate of missing values (25.7% in AQI and 13.2% in AQI-36). Along with Yi et al. (2016), we consider as the test set the months of March, June, September and December. We consider both the in-sample and out-of-sample scenarios. In latter case, we do not consider windows overlapping with any of the test months. We use the same procedure of Yi et al. (2016) to simulate the presence of missing data for evaluation.

---

[2] https://github.com/nnzhan/Graph-WaveNet
[3] https://github.com/caow13/BRITS
[4] https://github.com/Graph-Machine-Learning-Group/grin

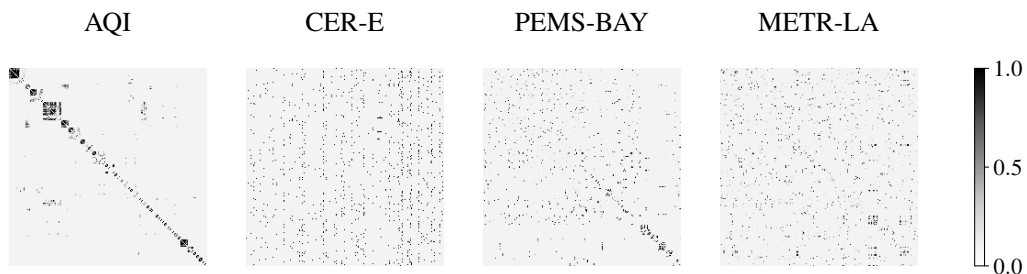

Figure 4: Adjacency matrices of the different datasets.

We select windows of data of length $T = 24$ for AQI and $T = 36$ for AQI-36 (in line with Cao et al. (2018)). To evaluate the imputation performances, we mask out from the test set and use as ground-truth the value $x_t^{i,j}$ if: (1) the value is not missing ($m_t^{i,j} = 1$) and (2) the value is missing at the same hour and day in the following month. Besides air quality readings, the dataset provides geographic coordinates of each monitoring station. To obtain an adjacency matrix from the geographic distances between nodes, we use a thresholded Gaussian kernel (Shuman et al., 2013): the weight $w_t^{i,j} = w^{i,j}$ of the edge connecting $i$-th and $j$-th node is

$$w^{i,j} = \begin{cases} \exp\left(-\frac{\text{dist}(i,j)^2}{\gamma}\right) & \text{dist}(i,j) \leq \delta \\ 0 & \text{otherwise} \end{cases}, \tag{12}$$

where $\text{dist}(\cdot, \cdot)$ is the geographical distance operator, $\gamma$ controls the width of the kernel and $\delta$ is the threshold. We set $\gamma$ to the standard deviation of geographical distances in AQI-36 in both datasets. We set $\delta$ so that it corresponds to a distance of $\approx 40$ km.

## B.2 TRAFFIC

The study of traffic networks is key for the development of intelligent transportation systems and a relevant application field for network science. While previous works (Yu et al., 2017; Li et al., 2018; Wu et al., 2019; Shang et al., 2020) have assessed spatio-temporal deep learning methods for the traffic forecasting task, we focus on reconstruction. We use as benchmark the *PEMS-BAY* and *METR-LA* datasets from Li et al. (2018). PEMS-BAY contains 6 months of data from 325 traffic sensors in San Francisco Bay Area, while METR-LA contains 4 months of sensor readings from 207 detectors in the Los Angeles County Highway (Jagadish et al., 2014); for both datasets, the sampling rate corresponds to 5 minutes.

We use input sequences of 24 steps, which correspond to 2 hours of data. For adjacency, we use a thresholded Gaussian kernel applied to geographic distances following previous works Wu et al. (2019). We split the data into three folds, using 70% of them for training and the remaining 10% and 20% for validation and testing, respectively.

## B.3 SMART GRIDS

We consider data from the Irish Commission for Energy Regulation (CER) Smart Metering Project[5]. We select only the subset of the available smart meters monitoring energy consumption of small and medium-sized enterprises (SMEs), i.e., 485 time series with samples acquired every 30 minutes. Note that access to dataset can be obtained free of charge for research purposes.

We build an adjacency matrix by extracting a k-nearest neighbor graph (with $k = 10$) from the similarity matrix built by computing the week-wise correntropy (Liu et al., 2007) among time series. As in the traffic case, we use a 70%/10%/20% split for training, validation and testing and use a window size of 24 steps. Data were normalized using standard scaling as in the previous settings, and we did not perform additional preprocessing steps.

---

[5]https://www.ucd.ie/issda/data/commissionforenergyregulationcer

## C  Additional results

In this appendix, we show an additional experiment in a controlled environment, comparison against additional baselines, additional ablation studies, and sensitivity analyses.

### C.1  Synthetic data

In this experiment, we test our method on the simulated particle system introduced by Kipf et al. (2018)[6]. We simulate the trajectories of $N = 10$ particles in a $(10 \times 10)$ box with elastic collision. Each particle carries a either positive or negative charge $q \in \{1, -1\}$. Two particles attract each other if they have opposite sign, otherwise they repel. Interaction forces between two particles are ruled by Coulomb's law. We collect two datasets, each containing 5000 independent simulations of $T = 36$ steps each. In the first dataset, particles have the same charge in every simulation. In the second one, we sample the charges uniformly at random at the beginning of every simulation. In both scenarios, the initial location and velocity of the particles are drawn randomly. At each step, we randomly remove blocks of consecutive readings with a probability $p_{failure} = 2.5\%$ and a length sampled uniformly from the interval $[4, 9]$. Here, a reading consists of the $(x, y)$ coordinates of the particle's position. We further mask out $2.5\%$ of positions at random. The percentage of values not masked out is $\approx 74\%$. For evaluation purposes, we generate another mask using the same missing data distribution and use the masked values as ground-truth for evaluation. We split dataset in training/validation/test folds using $70\%/10\%/20\%$ splits, respectively.

Table 6: Results on the synthetic datasets. Performance averaged over 5 runs.

| Model | Fixed charge | | Varying charge | |
|---|---|---|---|---|
| | MAE | MSE | MAE | MSE |
| BRITS | $0.1203 \pm 0.0003$ | $0.0878 \pm 0.0002$ | $0.1089 \pm 0.0007$ | $0.0840 \pm 0.0001$ |
| **GRIN** | $\mathbf{0.0500} \pm \mathbf{0.0055}$ | $\mathbf{0.0061} \pm \mathbf{0.0010}$ | $\mathbf{0.0530} \pm \mathbf{0.0092}$ | $\mathbf{0.0074} \pm \mathbf{0.0033}$ |
| Improv. | $2.41\times$ | $14.39\times$ | $2.05\times$ | $11.35\times$ |

We test our method (GRIN) and BRITS in both synthetic datasets. We use 32 units for the hidden layer of BRITS ($\approx 25K$ parameters) and 16 units for both the encoder and decoder of GRIN ($\approx 10K$ parameters). Results are reported in Tab. 6. Both the methods take as input only the particles' positions, with no information about the charges. As can be seen, consistently with what observed by Kipf et al. (2018), relational representations are impressively effective in this scenario. Our method outperforms the baseline by more than an order of magnitude in terms of MSE. Surprisingly, BRITS is more accurate in the setting with varying charge. Our hypothesis is that the added stochasticity acts as a regularization and forces BRITS to learn a more general model.

### C.2  Empirical comparison against matrix factorization with side information

Table 7: Comparison of regularized matrix factorization methods on air quality datasets. Results averaged over 5 independent runs.

| Model | AQI-36 | | | AQI | | |
|---|---|---|---|---|---|---|
| | MAE | MSE | MRE (%) | MAE | MSE | MRE (%) |
| MF | $30.54 \pm 0.26$ | $2763.06 \pm 63.35$ | $43.84 \pm 0.38$ | $26.74 \pm 0.24$ | $2021.44 \pm 27.98$ | $40.01 \pm 0.35$ |
| GRMF | $19.29 \pm 0.39$ | $1054.48 \pm 40.79$ | $27.68 \pm 0.56$ | $26.38 \pm 0.32$ | $2031.21 \pm 72.10$ | $39.48 \pm 0.48$ |
| TRMF | $15.97 \pm 0.14$ | $1178.65 \pm 60.14$ | $22.92 \pm 0.20$ | $21.86 \pm 0.28$ | $1516.81 \pm 45.53$ | $32.71 \pm 0.42$ |
| **GRIN** | $\mathbf{10.51} \pm \mathbf{0.28}$ | $\mathbf{371.47} \pm \mathbf{17.38}$ | $\mathbf{15.09} \pm \mathbf{0.40}$ | $\mathbf{13.10} \pm \mathbf{0.08}$ | $\mathbf{615.80} \pm \mathbf{10.09}$ | $\mathbf{19.60} \pm \mathbf{0.11}$ |

As mentioned in Section 2, several matrix factorization approaches – often studied in the context of recommender systems – can be regularized by considering priors on the spatio-temporal struc-

---

[6]https://github.com/ethanfetaya/NRI

ture of the data. Intuitively, spatial regularization is achieved by imposing soft constraints on the smoothness of the interpolated function w.r.t. nodes of an underlying graph (Cai et al., 2010; Rao et al., 2015). Temporal regularization can be obtained by imposing analogous constraints modelling temporal dependencies as – eventually weighted – edges of a graph. In temporal regularized matrix factorization (TRMF; Yu et al., 2016), similarly, coefficients of an autoregressive model are used as temporal regularizer.

Tab. 7 shows a comparison of different matrix factorization approaches on imputation in the air quality datasets (where we considered the in-sample setting in Section 5). For TRMF we used an implementation adapted from the Transdim repository[7], while for graph regularized matrix factorization (GMRF) we use a custom implementation of the method proposed by Cai et al. (2010). We fixed the rank to be equal to 10 (as the one used in all the experiments for standard MF) and tuned the regularization coefficients on a validation set. Results do show that introducing spatial and temporal regularization improve w.r.t. vanilla MF; however, deep learning methods – and even linear VAR predictors – achieve far superior reconstruction accuracy here. Arguably, low-rank approximation methods might instead have an edge in a low-data regime: this type of analysis is, however, out of the scope of this work.

## C.3 SCALABILITY

With reference to a standard bidirectional GRU, using MPNNs to implement the cell's gates increases the computational complexity by a factor that scales with the number of edges $O(E)$ – if using an efficient sparse implementation – or with the number of nodes squared $O(N^2)$. Luckily, this overhead can be amortized as most of the computation can be parallelized. Research on scalable and memory efficient GNNs is a very active field (e.g., Hamilton et al., 2017; Frasca et al., 2020): depending on the task, the designer can opt for massage passing operators that meet the application requirements in terms of performance, time and space constraints.

## C.4 ABLATION STUDY

Here we provide two different ablation studies, the first one on the architecture of GRIN and the second one on the graph structure.

### C.4.1 ARCHITECTURAL ABLATIONS

Tab. 8 shows additional results for the ablation study presented in Section 5. Consistently with what we already observed, the spatial decoder and bidirectional architecture improve performance and appear particularly relevant in settings with blocks of missing data.

Table 8: Ablation study. MAE averaged over 5 runs. (P) and (B) indicate the Point Missing and Block Missing settings, respectively.

| Model | AQI | METR-LA (B) | METR-LA (P) | CER-E (B) | CER-E (P) |
|---|---|---|---|---|---|
| **GRIN** | **14.73** $_{\pm\,0.15}$ | **2.03** $_{\pm\,0.00}$ | **1.91** $_{\pm\,0.00}$ | **0.42** $_{\pm\,0.00}$ | **0.29** $_{\pm\,0.00}$ |
| w/o sp. dec. | 15.40 $_{\pm\,0.14}$ | 2.32 $_{\pm\,0.01}$ | 2.01 $_{\pm\,0.00}$ | 0.49 $_{\pm\,0.00}$ | **0.29** $_{\pm\,0.00}$ |
| w/ denoise dec. | 17.23 $_{\pm\,1.12}$ | 2.96 $_{\pm\,0.18}$ | 2.09 $_{\pm\,0.02}$ | 0.61 $_{\pm\,0.02}$ | 0.32 $_{\pm\,0.00}$ |
| MPGRU | 18.76 $_{\pm\,0.11}$ | 2.57 $_{\pm\,0.01}$ | 2.44 $_{\pm\,0.00}$ | 0.53 $_{\pm\,0.00}$ | 0.41 $_{\pm\,0.00}$ |

### C.4.2 GRAPH STRUCTURE ABLATIONS

Here we study how exploiting the relational structure of the problem affects the accuracy of the reconstruction. In particular, we run two additional experiments on the METR-LA dataset (Block missing settings), where instead of using as adjacency matrix the thresholded kernel in Eq. 12, we use (1) a fully connected graph ($W = \mathbb{1}$) and (2) a graph with no edges ($W = \mathbb{I}$). To provide node – i.e., sensor – identification, we use learnable embeddings as additional node features. Results are

---

[7] https://github.com/xinychen/transdim

shown in Tab. 9, performance for BRITS are reported as reference. It is clear that the constraints posed by the graph structure do have an impact on the accuracy of missing data imputation and, at the same time, that spatial information is relevant for the task.

Table 9: Performance with different adjacency matrices. Results averaged over 5 runs. (B) indicates the Block Missing setting.

| | METR-LA (B) | | |
|---|---|---|---|
| Method | MAE | MSE | MRE (%) |
| **GRIN** | **2.03** $\pm$ **0.00** | **13.26** $\pm$ **0.05** | **3.52** $\pm$ **0.01** |
| fully connected | 2.63 $\pm$ 0.01 | 27.37 $\pm$ 0.38 | 4.56 $\pm$ 0.02 |
| no edges | 3.42 $\pm$ 0.04 | 51.68 $\pm$ 0.71 | 5.93 $\pm$ 0.08 |
| BRITS | 2.34 $\pm$ 0.01 | 17.00 $\pm$ 0.14 | 4.05 $\pm$ 0.01 |

## C.5 SENSITIVITY ANALYSIS

Finally, in this subsection we carry out an assessment of performance degradation w.r.t. the amount of missing data. Before discussing results, there are a few remarks that are worth bringing up regarding imputation in highly sparse settings. In the first place, GRIN, as well as a large portion of the state-of-the-art baselines, is an autoregressive model, which means that it might be subject to error accumulation over long time horizons. Furthermore, here, consistently with Section 5, we consider the out-of-sample setting which is particularly challenging in the sparse data regime. That being said, GRIN achieves remarkable performance also in this benchmark.

We train one model each for GRIN and BRITS by randomly masking out 60% of input data for each batch during training, then, we run the models on the test set by using evaluation masks with increasing sparsity (note that this causes a distribution shift in evaluation). For each level of sparsity, evaluation is repeated 5 times by sampling different evaluation masks. Results are reported in Tab. 10 and Fig. 5 shows that GRIN outperforms BRITS in all the considered scenarios.

Table 10: Performance with different amounts of missing data. Results averaged over 5 different evaluation masks in the out-sample setting. (P) indicates the Point Missing setting.

| | METR-LA (P) | | | | | | | | |
|---|---|---|---|---|---|---|---|---|---|
| % Missing | 10 | 20 | 30 | 40 | 50 | 60 | 70 | 80 | 90 |
| **GRIN** | **1.87** $\pm$ **0.01** | **1.90** $\pm$ **0.00** | **1.94** $\pm$ **0.00** | **1.98** $\pm$ **0.00** | **2.04** $\pm$ **0.00** | **2.11** $\pm$ **0.00** | **2.22** $\pm$ **0.00** | **2.40** $\pm$ **0.00** | **2.84** $\pm$ **0.00** |
| BRITS | 2.32 $\pm$ 0.01 | 2.34 $\pm$ 0.00 | 2.36 $\pm$ 0.00 | 2.40 $\pm$ 0.00 | 2.47 $\pm$ 0.00 | 2.57 $\pm$ 0.01 | 2.76 $\pm$ 0.00 | 3.08 $\pm$ 0.00 | 4.02 $\pm$ 0.01 |

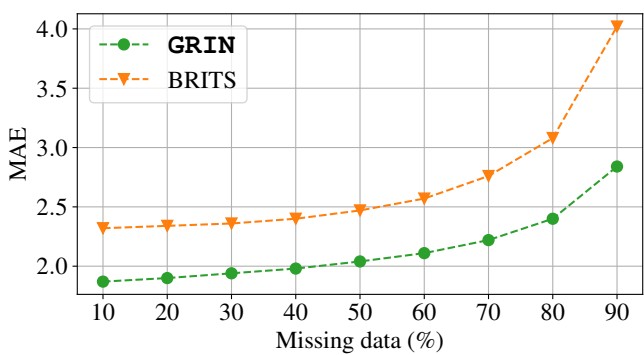

Figure 5: The plot shows graphically the results in Tab. 10.

