# OpenReview forum: "Filling the G_ap_s: Multivariate Time Series Imputation by Graph Neural Networks"
_ICLR.cc/2022/Conference — ICLR 2022 Poster_

### Official Review · Reviewer_9yUs · 2021-10-26

**Correctness:** 4
**Technical Novelty And Significance:** 2
**Empirical Novelty And Significance:** 2
**Recommendation:** 6
**Confidence:** 2

**Main Review:**

The paper is well written, and easy to understand. The problem is well motivated, and I didn't see any gaps in the solution presented.

I found it interesting that spatio-temporal methods to impute time series values do not take graphs into account. Nonetheless, there are methods that do model multivariate time series as low rank matrices, and use matrix factorization style methods for imputation. I recommend the authors mention this and compare and contrast to these methods.

In the related work section , you mention a bunch of methods that use graphs to forecast time series.  Why can't these be used for imputation? This also raises another question: If one can forecast in the presence of missing values, is imputation necessary?

I'm curious about the reason to incorporate backward dynamics. Unlike text, time only moves "forward". However, table 3 shows that bidirectionality helps. Can you provide some intuition for this?

How scalable is the method? GRUs themselves are probably scalable, but adding message passing in the gates adds overhead. I would want to see a memory/runtime comparison to the baseline methods on the datasets. The datasets used themselves are not too large, so maybe even a simulation or some justification might be good.

What happens if all 'd' features are missing for a particular node? Is the entire feature vector imputed? It might be good to have a study on a toy dataset where you examine various "types" of missing data, varying the length and randomness of missing features and compare your method to some of the baselines. I think this will make the paper far more compelling.

**Summary Of The Paper:**

In this paper, the authors propose a GNN based imputation method for Time Series.  They argue that past works for time series imputation only take time into account, and not the spacial relationships between time series. The model (called GRIN) works by encoding the data via a GNN, and performing 2 rounds of imputation. The GNN itself acts as gates in a GRU that accounts for the sequential information. This is repeated for a "forward" and "backward" pass, and a final MLP infers the missing value. Experiments on multiple datasets show that the proposed method outperforms baselines.

**Summary Of The Review:**

There are a few aspects I'm not completely sure of, hence my rating and confidence level:

1. Time series imputation itself is sometimes motivated from the point of view of making good predictions. To this end, the authors might want to motivate this aspect a bit more, and also compare to methods that can make predictions from incomplete data. I don't see that addressed in the paper at all. If this is a completely orthogonal problem, that is something that the authors should also clarify.

2. I'm not sure how scalable the method is, which is of importance. Multivariate time series can often get quite large, and the authors should address scalability either via a computational complexity argument, or some empirical evidence.

3. combining time series and graphs is not new. There has been work in the context of forecasting (like the authors mention) and some other papers that have not been cited. The major contribution in this work seems to be adding graphs to a "GRU". But this kind of RNN + graph architecture has also been considered before.

---

> ### Author Response · Authors · 2021-11-12
> **Response to Reviewer 9yUs comments (part I)**
>
> Thank you for your interesting comments and your positive opinion about our work.
>
> > Nonetheless, there are methods that do model multivariate time series as low rank matrices, and use matrix factorization style methods for imputation. I recommend the authors mention this and compare and contrast to these methods.
>
> We will mention this line of work and discuss the relationship with our work in the revision of the paper.
>
> > In the related work section, you mention a bunch of methods that use graphs to forecast time series. Why can't these be used for imputation? This also raises another question: If one can forecast in the presence of missing values, is imputation necessary?
>
> Sure, they can be used for imputation ( MPGRU is an example of that), but using one-step-ahead predictors to perform imputation does not lead to very good results as shown in the ablation study and by the results achieved by MPGRU. Previous work on forecasting often used a simple interpolation strategy for filling up missing data and assumed that there were no missing values in the input time series (and, thus, relied on an imputation preprocessing step). Furthermore, imputation can be an objective on its own even without considering another downstream task.
>
> > I'm curious about the reason to incorporate backward dynamics. Unlike text, time only moves "forward". However, table 3 shows that bidirectionality helps. Can you provide some intuition for this?
>
> The problem is that even if time moves forward, in general, we do not have access to the full state of the system, but only to noisy observations (through sensors). The backward information helps in disambiguating the true (latent) state of the system and, thus, reconstructing missing observations.
>
> > How scalable is the method? GRUs themselves are probably scalable, but adding message passing in the gates adds overhead. I would want to see a memory/runtime comparison to the baseline methods on the datasets. The datasets used themselves are not too large, so maybe even a simulation or some justification might be good.
>
> This is a good point. The scalability of graph neural networks is a hot research topic and certainly something to keep in mind when using these approaches. See our comments down below.
>
> > What happens if all 'd' features are missing for a particular node? Is the entire feature vector imputed?
>
> Yes, here we considered datasets with only one input dimension for each node. In the general d-dimensional case, all (and only) the missing features would be reconstructed.
>
> > It might be good to have a study on a toy dataset where you examine various "types" of missing data, varying the length and randomness of missing features and compare your method to some of the baselines. I think this will make the paper far more compelling.
>
> We will add a comparison of BRITS and GRIN with an increasing number of missing data in the appendix. However, we'd like to point out that in the AQI missing data already follow several different patterns (e.g., in 2% of time-steps no data is available at any sensor).
>
> > Time series imputation itself is sometimes motivated from the point of view of making good predictions. To this end, the authors might want to motivate this aspect a bit more, and also compare to methods that can make predictions from incomplete data. I don't see that addressed in the paper at all. If this is a completely orthogonal problem, that is something that the authors should also clarify.
>
> The reasons for which one might be interested in doing imputation are manifold, one of them, as you mention, is as a preprocessing step for a downstream task (e.g. forecasting or time-series classification). Here we decided to be agnostic to the downstream task and focus on imputation itself. It should be noted that, however, most existing state-of-the-art forecasting methods do assume that input data do not have missing values.
>
> > I'm not sure how scalable the method is, which is of importance. Multivariate time series can often get quite large, and the authors should address scalability either via a computational complexity argument, or some empirical evidence.
>
> We will add a comment on the computational requirements on the revision of the paper. In general, the computational complexity required to perform message passing scales with the number of edges. Our architecture was not designed with scalability in mind (however the bidirectional model can be easily trained with a batch size of 32 on a GPU with <8GB of memory for sensors networks with hundreds of sensors (>400). A more scalable architecture could be designed by exploiting recent advances in applying GNNs to very large graphs.

---

> ### Author Response · Authors · 2021-11-12
> **Response to Reviewer 9yUs comments (part II)**
>
> > Combining time series and graphs is not new. There has been work in the context of forecasting (like the authors mention) and some other papers that have not been cited. The major contribution in this work seems to be adding graphs to a "GRU". But this kind of RNN + graph architecture has also been considered before.
>
> Certainly combining GNNs and recurrent architectures have already been considered and we cite several works which addressed this aspect (e.g., [1, 2] ), however, this is not our main contribution. Our main focus is, instead, the study of an effective graph-based framework for multivariate time series imputation.
>
> [1] Li, Y., Yu, R., Shahabi, C., & Liu, Y., Diffusion Convolutional Recurrent Neural Network: Data-Driven Traffic Forecasting. ICLR 2021.
>
> [2] Seo, Y., Defferrard, M., Vandergheynst, P., & Bresson, X., Structured sequence modeling with graph convolutional recurrent networks. NeurIPS 2018.

---

### Official Review · Reviewer_jZvK · 2021-10-26

**Correctness:** 3
**Technical Novelty And Significance:** 3
**Empirical Novelty And Significance:** 3
**Recommendation:** 6
**Confidence:** 3

**Main Review:**

Novelty:

I think this is the first work that takes available relational information into consideration when building the imputation model, and the utilization of GNN makes sense.

Clarity:

What do you mean by "relational information"? Is that human-annotated information that, for example, describes which sensors are related? Or is that not provided by humans but automatically learned by the model?

Section 4 is a little bit hard for me to follow to judge the novelty of the proposed architecture. The spatial information is already considered in the first stage as shown in equation 2 (message passing from neighbors), why do we need a second stage that also considers spatial information?  In the "Unidirectional model" paragraph of section 4, $H_{[t,t+T]} \in R^{N_t \times l}$, what is $l$? In equation 9, the term $h^i_{t-i}$ is really confusing for me, why do you subtract node index i from t?

Limitation of the approach:

The reliance on RNNs makes the proposed approach unable to handle irregularly sampled data. Also, this paper assumes a static adjacency matrix, which makes it not suitable for dynamic relational data.

Experiment:

The extensive comparison to BRITS on different datasets clearly shows the advantage of the proposed approach. However, my concern is that the way that the missing data are simulated is too simple: block missing only drops 5% of the data, and point missing only masks out 25% of the data. I am not convinced that the proposed method can work well when the observation becomes very sparse. I suggest increasing the number of missing data and compare to BRITS.

**Summary Of The Paper:**

This paper proposes to leverage GNN that takes available relational information for multivariate time series imputation. Each feature dimension of the time series is regarded as a node in GNN and the message-passing is used to implement the update function of bidirectional RNN. On three benchmarks, this new method achieves better performance than the previous STOA method BRITS.

**Summary Of The Review:**

My score is 5 for now as there are some clarity issues. But I am happy to increase my score if the authors can answer these questions.

---

> ### Author Response · Authors · 2021-11-12
> **Response to Reviewer jZvK comments**
>
> Thank you for your insightful comments, we will do our best to clarify your doubts about our work.
>
> > What do you mean by "relational information"? Is that human-annotated information that, for example, describes which sensors are related? Or is that not provided by humans but automatically learned by the model?
>
> In general, with relational information, we refer to any functional dependency that might exist between sensors, e.g., as you said, by describing how sensors are correlated. Here, we consider both settings where the information is already available (e.g., geographical information) or is extracted from data (e.g., by using a similarity score). Future work could exploit end-to-end solutions to learn at the same time the underlying structure and the imputation model.
>
> > The spatial information is already considered in the first stage as shown in equation 2 (message passing from neighbors), why do we need a second stage that also considers spatial information?
>
> The spatial decoder aggregates spatial information at time step t which was not available at previous time steps, note that this step can be critical for good performance (as shown in the ablation study). Here the model learns to reconstruct observed values only by looking at neighboring nodes (see Eq. 9). We are available for more clarifications if needed.
>
> > In the "Unidirectional model" paragraph of section 4,  H[t,t+T]∈RNt×l, what is l ? In equation 9, the term ht−ii is really confusing for me, why do you subtract node index i from t?
>
> "l" is the dimension of the hidden representation. Sorry, that was a typo, the “i” was supposed to be a “1”.
>
> > The reliance on RNNs makes the proposed approach unable to handle irregularly sampled data. Also, this paper assumes a static adjacency matrix, which makes it not suitable for dynamic relational data.
>
> RNNs that handle irregularly sampled time series have been studied in the literature (see GRU-D[1] and GRUI[2]), and exploiting our approach in that context could be an interesting extension (which is beyond the scope of our work at the moment). We have a similar position regarding dynamic graphs, extension of our approach to handle them is a straightforward and interesting direction for future work (e.g., one might incorporate a model to learn a different adjacency matrix for each window of data as in NRI [3]).
>
> > The extensive comparison to BRITS on different datasets clearly shows the advantage of the proposed approach. However, my concern is that the way that the missing data are simulated is too simple: block missing only drops 5% of the data, and point missing only masks out 25% of the data. I am not convinced that the proposed method can work well when the observation becomes very sparse. I suggest increasing the number of missing data and compare to BRITS.
>
> In the experiments, we focused on settings with a realistic percentage of missing data. However, we are running an experiment with an increased percentage of missing values for the appendix of the paper to be added in the revision of the paper.
>
> > My score is 5 for now as there are some clarity issues. But I am happy to increase my score if the authors can answer these questions.
>
> We hope that our comments helped in clarifying your doubts, we are available for further discussion.
>
> [1] Che, Z., Purushotham, S., Cho, K., Sontag, D., & Liu, Y., Recurrent neural networks for multivariate time series with missing values. Scientific reports, 2018.
>
> [2] Luo, Y., Cai, X., Zhang, Y., Xu, J., & Yuan, X., Multivariate time series imputation with generative adversarial networks. NeurIPS 2018
>
> [3] Kipf, T., Fetaya, E., Wang, K. C., Welling, M., & Zemel, R., Neural relational inference for interacting systems. ICML 2018

---

> > ### Comment · Reviewer_jZvK · 2021-11-19
> > **Most of my concerens are addressed and I increase my score to 6.**
> >
> > Most of my concerens are addressed and I increase my score to 6.

---

> > > ### Author Response · Authors · 2021-11-19
> > > **Thank you for the feedback**
> > >
> > > Thank you for the positive feedback and for your contribution to improving our work. We just uploaded a revision of the paper further addressing some of the points you raised.

---

### Official Review · Reviewer_JPtB · 2021-10-31

**Correctness:** 3
**Technical Novelty And Significance:** 3
**Empirical Novelty And Significance:** 2
**Recommendation:** 8
**Confidence:** 4

**Main Review:**

The paper tackles an important and general problem for numerous application areas where time-series data from multiple sensors is processed with missing data (as can be expected in reality). The architecture builds on previous work from graph RNN approaches and message passing neural networks, as is clearly presented in the paper. The emphasis is to include the spatial information of the problem into the architecture, this inductive bias can then help in imputation as the neighbourhood of each missing data point can help in accurately estimating the missing value. This paper is clearly written and covers a lot of related work. The ideas bring the graph structure into the problem setting and the results are impressive.

In the experiments, BRITS is a state-of-the-art approach, but can you include more state-of-the-art approaches for comparison? For example NAOMI [Liu et al., 2019], E2GAN [Luo et al, 2019]. Additionally, kNN is a good baseline comparison, as it uses the neighbourhood information, other good baselines that use neighbourhood information could be MF with graph side information, e.g. [Rao et al., 2015] or similar, to replace simple MF, and krigging that is a simple popular approach. One last comparison that may be relevant is to another method that uses both spatial and temporal information for MTSI [Yi et al, 2016].

This architecture seems to share similarities with other RNN like networks that work with graph structures, but do not include imputation, can you compare with this line of work, for example [Li at al., 2016]?

Yujia Li, Daniel Tarlow, Marc Brockschmidt, Richard S. Zemel: Gated Graph Sequence Neural Networks. ICLR (Poster) 2016

Rao, Nikhil, et al. "Collaborative Filtering with Graph Information: Consistency and Scalable Methods." NIPS. Vol. 2. No. 4. 2015.

Yi, Xiuwen, et al. "ST-MVL: filling missing values in geo-sensory time series data." (2016).

Luo, Yonghong, et al. "E2gan: End-to-end generative adversarial network for multivariate time series imputation." AAAI Press. 2019.


**Summary Of The Paper:**

A novel recurrant graph neural network, for spatio-temporal encoding, coupled with a message passing neural network, for spatial decoding, is proposed for imputing missing values in a multivariate time series; The architecture uses temporal and spatial properties of the data. It seems that the approach of gated recurrent neural networks (and related architectures using gated recurrent units, GRUs) is extended to include the graph topology through neighbourhood message passing. Experiments show impressive results on three benchmark datasets, and comparison with simple and state-of-the-art baselines. A short ablation study excludes the spatial decoder and omits the bi-directional architecture to empricially show that they have a significant impact on the imputation performance. Another short experiment tests the ability of this architecture to replace entire missing sensors, using an air pollution dataset with multiple sensors, results are convincing.

**Summary Of The Review:**

A novel contribution that brings the graph structure more strongly into the architecture than previously, building on existing work in the domain. Evaluated on relevant benchmark datasets (although one or two more could be added) against benchmark methods (again where one or two more could be added) and ablation studies included to give some insight to the impact of the contributions on the qulaity of the estimations of the missing data.

---

> ### Author Response · Authors · 2021-11-12
> **Response to Reviewer JPtB comments**
>
> Thank you for your comments and the interest in our work, they are both really appreciated.
>
> > In the experiments, BRITS is a state-of-the-art approach, but can you include more state-of-the-art approaches for comparison? For example NAOMI [Liu et al., 2019], E2GAN [Luo et al, 2019]. Additionally, kNN is a good baseline comparison, as it uses the neighbourhood information, other good baselines that use neighbourhood information could be MF with graph side information, e.g. [Rao et al., 2015] or similar, to replace simple MF, and krigging that is a simple popular approach. One last comparison that may be relevant is to another method that uses both spatial and temporal information for MTSI [Yi et al, 2016].
>
> We would like to point out that the included rGAIN (according to the recent literature) is a very competitive baseline. As indicated in the paper, the model corresponds to the unsupervised version of SSGAN [1], which outperforms NAOMI and BRITS by a good margin in some of the settings studied by Miao et al.. NAOMI would not be directly applicable in our setting since they only consider the case where, at each time step, either all data are available or are completely missing. Clearly, this assumption makes their model very difficult to scale in our setting (one should apply their recursive imputation strategy for each missing value in each channel). We actually ran experiments with E^2GAN, but the results were disappointing and we did not include them in the paper to dedicate more computational power to stronger baselines. Regarding STMVL, it works specifically on geotagged series and is outperformed by the more recent baselines already included in the paper (e.g., BRITS).  For what concerns MF with graph side information, it is indeed a relevant line of research (which however could be used only in the "in sample" setting), we will try to add a baseline in this direction compatibly with the time constraints. If you are interested in discussing any of these aspects in more detail, please do add a follow-up comment and we will be happy to answer and give you more details.
>
> > This architecture seems to share similarities with other RNN like networks that work with graph structures, but do not include imputation, can you compare with this line of work, for example [Li at al., 2016]?
>
> This is an interesting point, however, the architecture introduced by Li et al. while being recurrent does not account for temporal data, but applies a recurrent cell to propagate information in the graph structure by aggregating messages coming from wider and wider neighborhoods. That being said, we will add a comment on the similarities shared with our architecture, together with already present references about recurrent GNNs for spatio-temporal data already cited in the paper. Furthermore, a simple MPGRU is already included as a baseline in the paper and several methods using recurrent GNNs are discussed in the related works (see DCRNN[2], GCRNN[3]).
>
> > A novel contribution that brings the graph structure more strongly into the architecture than previously, building on existing work in the domain. Evaluated on relevant benchmark datasets (although one or two more could be added) against benchmark methods (again where one or two more could be added) and ablation studies included to give some insight to the impact of the contributions on the qulaity of the estimations of the missing data.
>
> Regarding the additional experiments and baselines, the paper includes experiments on 7 benchmarks (5 + 2 in the appendix) with different settings for each dataset: we argue that these are enough to support our claims. Compatibly with the time constraints of the discussion period, we will try to add another baseline that models time series as low-rank matrices for the "in-sample" setting.
>
>
> [1] Miao, X., Wu, Y., Wang, J., Gao, Y., Mao, X., & Yin, J., Generative Semi-supervised Learning for Multivariate Time Series Imputation. AAAI 2021.
>
> [2] Li, Y., Yu, R., Shahabi, C., & Liu, Y., Diffusion Convolutional Recurrent Neural Network: Data-Driven Traffic Forecasting. ICLR 2021.
>
> [3] Seo, Y., Defferrard, M., Vandergheynst, P., & Bresson, X., Structured sequence modeling with graph convolutional recurrent networks. NeurIPS 2018.
>
> [4] Rao, N., Yu, H. F., Ravikumar, P., & Dhillon, I. S., Collaborative Filtering with Graph Information: Consistency and Scalable Methods. NeurIPS 2015.

---

### Official Review · Reviewer_hJvE · 2021-11-01

**Correctness:** 4
**Technical Novelty And Significance:** 3
**Empirical Novelty And Significance:** 3
**Recommendation:** 8
**Confidence:** 3

**Main Review:**

I really enjoyed reading this paper. The derivations are sound and it is very well written. I appreciate the readability of the methods section. The performance obtained in the experimental section are also very convincing.

- After reading this paper, I'm still not exactly sure as why the graph representation of the spatial correlations should lead to better performance. In 5.1, the authors state that "the results provide empirical evidence of the positive regularization effect of the inductive biases encoded into GRIN". I think the paper would benefit from a motivational or discussion section where these assumptions are more challenged and where authors could share their intuition and rationale behind this specific inductive bias.

- Regarding the above comment, it therefore seems legitimate to me to study the impact of different choices of graphs one can make. I would then appreciate greatly an experiment where the impact of the type of graph taken would be investigated. In particular, comparing against a fully connected graph (with the position of the nodes given in the input node features) and an identity matrix (as pointed to in section 3 - sequence of graphs).

- It's also not clear to me why you need two steps decoder. If I understand correctly, the. w/o spec decoder of table 3 corresponds to the architecture with the second step removed. Is it also possible to remove only the first imputation step (the linear one) ? Why is this linear step needed ?

- I really like section 5.2.

- You focus on spatio-temporal data. However, do the authors think that this method could be applicable to a broader class of time series where the different dimensions are not different locations ? For example, financial time series or healthcare time series ? If it's not the case , then I think the authors should be more clear about this assumption in the text.

- Equation 9 : I think the subscript of h is t-1 and not t-i


**Summary Of The Paper:**

This paper proposes a method for time series imputation modeling the spatial dependencies with graphs. The paper leverages a bi-directional temporal graph neural network architecture to learn node level representations that allows to impute missing values with better than state of the art accuracy.

**Summary Of The Review:**

I think it's a nice paper, that would spark a lot of discussion. Some more ablation studies are required to understand the relative importance of the different building blocks in the proposed architecture.

---

> ### Author Response · Authors · 2021-11-12
> **Response to Reviewer hJvE comments**
>
> Thank you for your comments, we are really glad that you enjoyed our paper. Here are our point-by-point answers to your comments.
>
> > After reading this paper, I'm still not exactly sure as why the graph representation of the spatial correlations should lead to better performance. [...] I think the paper would benefit from a motivational or discussion section where these assumptions are more challenged and where authors could share their intuition and rationale behind this specific inductive bias.
>
> We believe that the inductive biases coming from adopting graph neural networks in this context are positive since they restrict the space of models to plausible ones where spatial (relational) dependencies among sensors are explicitly accounted for. It is often the case, in fact, as the literature suggests, that functional dependencies among sensors play a crucial role in many inference tasks in sensor networks. Furthermore, the graph representation has a strong regularizing effect since it restricts node representations to depend on messages coming from neighboring nodes only, thus preventing overfitting useless information. Clearly, the downside is that there might be cases where such inductive bias might be wrong depending on the application domain. In the context of imputation, we like to think of the graph structure as imposing constraints to the values that can be observed at each node: functional dependencies among nodes make so that we can reconstruct missing values as shown, for example, in section 5.2. Thank you for the suggestion, we will improve the discussion on this aspect in the revision of the paper.
>
> > I would then appreciate greatly an experiment where the impact of the type of graph taken would be investigated. In particular, comparing against a fully connected graph (with the position of the nodes given in the input node features) and an identity matrix (as pointed to in section 3 - sequence of graphs).
>
> These are interesting points: using a fully connected graph and no graph at all would be interesting ablation studies. We are currently running experiments, we will write a general comment once we have results to show.
>
> > It's also not clear to me why you need two steps decoder. [...] Why is this linear step needed?
>
> We need the first decoding step (the linear one) in order to perform the message passing operations (spatial decoding) in Eq. (9), without the first decoding step, there will be no value associated with nodes with missing data at time step t (not all nodes have an associated valid x_t^j). Note that the spatial decoder uses hidden representations from t-1 and input values at time t.
>
> > You focus on spatio-temporal data. However, do the authors think that this method could be applicable to a broader class of time series where the different dimensions are not different locations ? For example, financial time series or healthcare time series ? If it's not the case , then I think the authors should be more clear about this assumption in the text.
>
> What we believe is the only requirement for our method to be applicable is the presence of any relational functional dependency among the channels of the time series. Furthermore, we do briefly mention in the paper, that a generic multivariate time series (where each channel might have a completely different meaning) could be represented as a heterogeneous graph (see Section 3). Extensions to heterogeneous graphs, as argued in the paper, should be straightforward and are indeed an interesting direction for future works/applications.
>
> > Equation 9 : I think the subscript of h is t-1 and not t-i
>
> Yes, thank you for noticing.

---

### Public Comment · ~Lijun_Sun1 · 2021-11-11
**missing previous works and baselines**

Multivariate time series imputation is an important research question. The presented research question (imputing missing values for signals generated from a graph structure) has been extensively studied in the transportation/traffic engineering domain (2/3 of the experiments are on traffic speed time series data), see e.g., Transdim (https://github.com/xinychen/transdim). I think this paper has missed a large body of research on this topic and the experiments are not very convincing.

I guess the virtual sensor experiment in 5.2 is equivalent to the graph signal interpolation and graph “Kriging” problem. There have been some works on this topic, which also utilize graph neural networks to achieve inductive power [1, 2].

The experiment only employs MF as a low-rank-based baseline. In fact, existing works have already leveraged graph information and temporal patterns [3, 4]. And temporal layers have been replaced by NN to accommodate nonlinearity. In fact, low-rank tensor models have been systematically examined for traffic data (e.g., Transdim). It is hard to say that the low-rank models do not perform well, as traffic time series data are in nature low-rank (e.g., the day-to-day similarity)

Regarding NN for MTSI, see some studies in the traffic time series context [e.g., the references in 5].

[1] Wu, Yuankai, Dingyi Zhuang, Aurelie Labbe, and Lijun Sun. "Inductive Graph Neural Networks for Spatiotemporal Kriging." In Proceedings of the AAAI Conference on Artificial Intelligence, vol. 35, no. 5, pp. 4478-4485. 2021.

[2] Appleby, Gabriel, Linfeng Liu, and Li-Ping Liu. "Kriging Convolutional Networks." In Proceedings of the AAAI Conference on Artificial Intelligence, vol. 34, no. 04, pp. 3187-3194. 2020.

[3] Rao, Nikhil, Hsiang-Fu Yu, Pradeep Ravikumar, and Inderjit S. Dhillon. "Collaborative Filtering with Graph Information: Consistency and Scalable Methods." In NIPS, vol. 2, no. 4, p. 7. 2015.

[4] Yu, Hsiang-Fu, Nikhil Rao, and Inderjit S. Dhillon. "Temporal regularized matrix factorization for high-dimensional time series prediction." Advances in neural information processing systems 29 (2016): 847-855.

[5] Liang, Yuebing, Zhan Zhao, and Lijun Sun. "Dynamic Spatiotemporal Graph Convolutional Neural Networks for Traffic Data Imputation with Complex Missing Patterns." arXiv preprint arXiv:2109.08357 (2021).

---

> ### Author Response · Authors · 2021-11-12
> **Response to public comment**
>
> Hi Lijun, thank you for your comment and interest in our work, and thank you for pointing us towards Transdim which is an interesting framework.
>
> Regarding references [1,2], they are indeed relevant to our work as kriging is definitely related to missing data imputation and we will make sure to reference them and comment on the relationship with our work. Thank you for pointing this out.
>
> As for MF methods with graph information, this was already pointed out by a reviewer, we will add comments on this line of work (which however is very different from our approach and often rely on different assumptions). You can also refer to the answers to reviewers' comments. We never said that low-rank models do not perform well. However, usually, these approaches work in the setting that we define as "in-sample" which makes up for only a small fraction of the experiments we performed, and all the traffic experiments (which are not the 2/3 of our experiments) were performed in the out-of-sample setting.
>
> For what concerns the time series forecasting methods developed in the context of traffic forecasting, there are already references in the paper, while in the experiments we focused on methods designed for imputation. In fact, regarding deep learning methods for general MTSI, we believe our review of previous work and the state of the art to be thorough. Sorry we missed [5], it came out on arxiv less than two weeks before the conference submission deadline.
>
> We, however, disagree with your comment about our experiments not being convincing: the commitment to a fair and thorough comparison of the different approaches was one of the guiding principles under which we performed our study and the code in the supplementary material is a testament to this.

---

### Author Response · Authors · 2021-11-19
**General comments and change list**

We thank again the reviewers for the all comments. Here you can find a detailed list of the changes we made in the revision of the paper, together with some comments. Thank you for helping us improve the quality of our work.

### Change list

* Added ablations on the graph structure.
   - Following the suggestion of Reviewer hJvE, we added an experiment to assess the impact of the graph structure (see Appendix C.4.2).
* Added sensitivity analysis.
    - As suggested by Reviewers jZvK and 9yUs, we added an experiment comparing BRITS and GRIN under several missing data rates. Results are reported in Appendix C.5.
* Added references to spatio-temporal low-rank approximation methods and added an appendix with experimental comparison on air quality imputation.
    - Besides mentioning matrix factorization methods with side information, we added an empirical comparison on the AQI dataset (which is the only experiment for which we consider the in-sample setting). We compare against MF methods with spatial and temporal regularization (see Appendix C.2), as suggested by reviewers JPtB and 9yUs.
* Added references to (spatio-temporal) kriging.
* Improved discussion of the motivation behind the approach.
* Added additional reference to different recurrent architectures for graph processing.
* Added appendix with discussion on scalability, see C.3.
* Corrected typos

---

> ### Author Response · Authors · 2021-11-22
> **Updated supplementary materials**
>
> We updated the supplementary materials to include the code for the additional matrix factorization baselines.

---

### Author Response · Authors · 2021-12-01
**Thank you**

Dear reviewers and AC,

since the rebuttal window is closing, we wish to thank all of you again for helping us improve our paper and for your knowledgeable comments.

We hope that our answers and revision of the paper helped in clearing up your doubts.

The authors

---

### Decision · Program_Chairs · 2022-01-20

**Decision:**

Accept (Poster)

**Comment:**

This paper proposes a method for time series imputation modelling the spatial dependencies with graphs, focusing on spatio-temporal data, where the spatial dimensions are represented by a graph.
The reviewers find the approach novel. The paper is well-written and clear. Related work is adequately discussed.
The experiments are convincing.
The reviewers agree that the paper should be accepted.